# Long-Term Follow-Up of Spinal Stenosis Inpatients Treated with Integrative Korean Medicine Treatment

**DOI:** 10.3390/jcm10010074

**Published:** 2020-12-28

**Authors:** Doori Kim, Joon-Shik Shin, Young-Joo Moon, Gwanghyun Ryu, Wonbin Shin, Jiyun Lee, Suyeon Lim, Hyun A Jeon, Ji-Yeon Seo, Wu Hao Wang, Jin-Ho Lee, Kyoung Sun Park, Yoon Jae Lee, In-Hyuk Ha

**Affiliations:** 1Jaseng Spine and Joint Research Institute, Jaseng Medical Foundation, Seoul 06110, Korea; doori.k07@gmail.com (D.K.); lovepks0116@gmail.com (K.S.P.); goodsmile8119@gmail.com (Y.J.L.); 2Jaseng Hospital of Korean Medicine, Seoul 06110, Korea; Jasengmaster@gmail.com (J.-S.S.); jasengjsr@gmail.com (J.-H.L.); 3Bucheon Jaseng Hospital of Korean Medicine, Bucheon 14598, Korea; duh2902@hanmail.net (Y.-J.M.); rayshine@naver.com (G.R.); swb3702@naver.com (W.S.); leejy5092@naver.com (J.L.); suyoun20@naver.com (S.L.); gusdk1218@naver.com (H.A.J.); wowpan21@gmail.com (J.-Y.S.); wuhao0117@naver.com (W.H.W.)

**Keywords:** integrative Korean medicine treatment, traditional Korean medicine, lumbar spinal stenosis, inpatients, back pain, acupuncture, herbal medicine, pharmacopuncture, Chuna manipulation

## Abstract

The present prospective observational study aimed to analyze the outcomes of inpatients who received integrative Korean medicine treatment in order to provide evidence on its effects on lumbar spinal stenosis (LSS). Patients with LSS who received inpatient treatment at four Korean medicine hospitals from January 2015 to December 2018 were followed up. Outcomes measured included the numeric rating scale (NRS) scores for back and leg pain, and Oswestry Disability Index (ODI). Changes in outcomes at admission, discharge, and follow-up, as well as associated predictors that could account for the improvement in outcomes were analyzed. The NRS score for back pain, NRS score for leg pain, and ODI decreased by 2.20 points (95% confidence interval (CI), −2.41 to −1.99), 2.28 points (95% CI, −2.59 to −1.96), and 17.31 points (95% CI, −19.6 to −15.02), respectively, at long-term follow-up compared with at admission. Patients with LSS who received inpatient integrative Korean medicine treatment exhibited an improvement in pain and functional disability. Further studies are required to determine the effects of integrative Korean medicine treatment.

## 1. Introduction

Lumbar spinal stenosis (LSS) is a degenerative disease in which the central canal is narrowed by the surrounding bones or tissues. Typical clinical symptoms of LSS include neurogenic claudication and radicular symptoms such as radicular pain in the lower extremities, numbness or tingling in the leg, hypoesthesia, and muscle weakness [1]. LSS has a prevalence rate of approximately 47% among individuals aged 60–69 years in the United States [2], and is the most common cause of spine surgery among those aged >65 years [3,4].

Surgical treatment for LSS is costly, carries a high adverse event risk, and often requires reoperation or readmission [5,6]. Furthermore, a standard treatment for LSS has not been established, and evidence on the effects of surgical treatment on LSS remains limited [7,8]. Weinstein et al. [1] and Atlas et al. [9] showed that the outcomes of patients who underwent surgery in the early stage were superior to those of patients who received conservative treatment. Nonetheless, Delitto et al. [10] reported no significant difference between these two types of treatment.

Conservative treatments for LSS include physical therapy, drugs (e.g., gabapentin), steroid injection, and acupuncture. In South Korea, a country with a dichotomized healthcare system [11,12], traditional Korean medicine treatments such as acupuncture, herbal medicine, and Chuna manipulation are widely employed as conservative treatments for LSS [13]. Despite this, only few related references are available. Among these are two case series that indicated the effectiveness of traditional Korean medicine treatment, including acupuncture [14,15]; a previous clinical study that reported a greater improvement in Oswestry Disability Index (ODI) in the acupuncture group than in the medication or exercise group [16]; and a randomized controlled trial (RCT) [17] enrolling 80 patients aged >50 years, which showed that the Roland Morris Disability Questionnaire score significantly decreased in the acupuncture group than in the sham acupuncture group. However, another RCT reported conflicting outcomes and showed that the acupuncture group did not exhibit a significant improvement in ODI, as compared to the usual care group [18]. One systematic review [19] did not identify conclusive evidence on the effectiveness and safety of acupuncture owing to methodological problems.

Basic research is therefore considered necessary in order to show the effects of Korean medicine treatment on LSS. Hence, the present study aimed to analyze the outcomes of inpatients who received integrative Korean medicine treatment in order to provide evidence on its effects on LSS. In this study, the effects of Korean medicine treatment on LSS were evaluated by conducting follow-ups on inpatients with LSS who received integrative Korean medicine treatment.

## 2. Materials and Methods

### 2.1. Study Design

The present prospective observational study was conducted on inpatients diagnosed with LSS and treated from January 2015 to December 2018 at four Korean medicine hospitals—namely Gangnam Jaseng Hospital of Korean Medicine, Bucheon Jaseng Hospital of Korean Medicine, Daejeon Jaseng Hospital of Korean Medicine, and Haeundae Jaseng Hospital of Korean Medicine. All of these four hospitals are designated by the Korean Ministry of Health and Welfare as specialized spine hospitals. A Korean medicine doctor diagnosed LSS by comprehensive review of patients’ symptoms, neurological examination, and imaging. A radiology specialist read the magnetic resonance imaging (MRI) scan.

Data on outcome measures of effects during the admission period and at follow-up were collected. In case of admission more than once within the period, the outcome measures at first admission were used. Korean medicine doctors conducted long-term follow-ups from June to August 2020 via a telephone survey, and outcome measures such as the numeric rating scale (NRS) score, ODI, and Patients’ Global Impression of Change (PGIC) scale score, as well as current treatment and treatment preferences, were surveyed. Internet survey questionnaires were distributed as text messages to patients who could not be reached after three call attempts, and questionnaires that were returned with answers were also used for analysis.

This study was conducted in accordance with the principles outlined in the 1975 Declaration of Helsinki. The study protocol was approved by the Institutional Review Boards of Jaseng Hospital of Korean Medicine (approval no.: JASENG 2020-05-010; approval date: 1 June 2020) and was registered at ClinicalTrials.gov (NCT02257723). Informed consent was obtained from all participants prior to their inclusion in the study.

### 2.2. Participants

The study participants were patients diagnosed with LSS and admitted to four Jaseng Hospitals of Korean Medicine across the country (Gangnam, Bucheon, Daejeon, and Haeundae) for integrative Korean medicine treatment from January 2015 to December 2018. With respect to the inclusion criteria, patients who (1) were admitted to Jaseng Hospital of Korean Medicine from January 2015 to December 2018 and diagnosed with LSS by a Korean medicine doctor; (2) were diagnosed with central stenosis based on MRI scan results; (3) had no communication problems; and (4) provided consent for their participation in the research were included in this study. As for the exclusion criteria, patients who (1) were diagnosed with a specific serious disease that could cause low back pain (e.g., tumor metastasis to the spine, acute fracture, spinal dislocation); (2) were admitted for pain due to a traffic accident; and (3) had a serious mental illness were excluded from the analysis. Other cases in which study participation was deemed difficult by the researcher were also excluded.

### 2.3. Intervention

Inpatients were treated according to the integrative Korean medicine treatment protocol. All treatments other than the protocol were allowed depending on patients’ condition and the judgment of the Korean medicine doctor, and all treatments during the admission period were recorded in electronic medical records. Integrative Korean medicine treatment included herbal medicine, acupuncture, pharmacopuncture, bee-venom pharmacopuncture, and Chuna manipulation.

#### 2.3.1. Herbal Medicine

Herbal medicine was administered as a 2-g pill or 120-mL water-based decoction 2–3 times/day. The main ingredients of the herbal medicine used in this study were *Saposhnikovia divaricata* Schiskin, *Achyranthis bidentata* Blume, *Acanthopanax sessiliflorum* Seem, *Cibotium barometz* J. Smith, *Glycine max* Merrill, and *Eucommia ulmoides* Oliver. The herbal medicine GCSB-5 (traditional name: Chungpa-Juhn) consists of the abovementioned medicinal ingredients, and in vivo and in vitro studies have investigated its anti-inflammatory [20], neuroprotective [21], and cartilage-protective [22] effects.

#### 2.3.2. Acupuncture

In acupuncture, a physician inserts disposable acupuncture needles (30 × 0.25 mm; Dong-bang Acupuncture, Seongnam, Korea) at Ah-shi points and related acupoints. In this study, acupuncture treatment was administered in 1–2 sessions/day. For one treatment, the needle retention time was approximately 15 min, and electrical stimulation was applied during acupuncture.

#### 2.3.3. Pharmacopuncture

The ingredients of pharmacopuncture are similar to those of oral herbal medicine. The ingredients of herbal medicine were decocted and lyophilized and were subsequently mixed with the prepared powder and normal saline. These were concurrently conducted at the time of acupuncture 1–2 times/day, and pharmacopuncture ingredients (>1 cm^3^) were injected each time using a syringe (CPL, 1 cm^3^, 26 G × 1.5 syringe; Shinchang Medical Co., Gumi, Korea) around the waist area.

#### 2.3.4. Bee-Venom Pharmacopuncture

Bee-venom pharmacopuncture was performed only when the result of a bee-venom test prior to the treatment was negative. Diluted bee-venom solution (mixed with normal saline at a ratio of 1000:1) was injected at 4–5 Ah-shi points at the physician’s discretion. Each point was injected with approximately 0.2 cm^3^ up to a total of 0.5–1.0 cm^3^ based on a Korean medicine doctor’s judgment using disposable injection needles (CPL, 1 cm^3^, 26 G × 1.5 syringe; Shinchang Medical Co., Gumi, Korea).

#### 2.3.5. Chuna Manipulation

Chuna manipulation refers to a semi-standardized manipulation used in traditional Korean medicine in which doctors use their hands, part of their body, or tools to stimulate the patients’ body structures. All physicians were trained through a standardized Chuna education course (3–5 sessions/week).

### 2.4. Outcome Measures

A Korean medicine doctor who had been trained in advance assessed all outcome measures at admission, discharge, and long-term follow-up. Sex, age, weight, height, smoking status, alcohol consumption, and medical history were included in the baseline assessment.

#### 2.4.1. Primary Outcome

##### NRS Score for Back Pain

The primary outcome of our study was the NRS score for back pain. The NRS is a numeric pain scale that objectively assesses the subjective pain felt by patients [23,24] and employs on an 11-point scale for the evaluation of current back pain, with “0” indicating no pain and “10” representing the worst pain imaginable. The NRS score was assessed at admission, discharge, and long-term follow-up.

#### 2.4.2. Secondary Outcomes

##### NRS Score for Leg Pain

The NRS score for leg pain was also measured at admission, discharge, and long-term follow-up.

##### ODI

Patients’ functional disability status was assessed using a 10-item ODI questionnaire developed for the evaluation of disability status in patients with low back pain [25]. Each question is divided into 6 levels (from 0 to 5 points); the higher the score, the more severe the degree of disability. A validated Korean ODI questionnaire [26] was used in this study, and ODI was measured at admission, discharge, and long-term follow-up.

##### Five-Level EuroQol 5-Dimension (EQ-5D-5L) Questionnaire

In this study, the five-level version of EuroQol 5-dimension (EQ-5D) questionnaire was used as a tool to evaluate the participants’ quality of life. EQ-5D is the most widely employed method for the indirect assessment of health-related quality of life. EQ-5D-5L comprises 5 items (mobility, self-care, usual activities, pain, anxiety/depression) including questions on current health status, with each question answered on a 5-point Likert scale (1 = “I have no problems about,” 2 = “I have slight problems about,” 3 = “I have moderate problems about,” 4 = “I have severe problems about,” 5 = “I am unable to about”). A validated Korean version of EQ-5D-5L [27] was used in this study.

##### Walking Time

The maximum walking time without pain was measured in minutes. Walking limitations are a hallmark of patients with LSS [28]; therefore, a number of LSS-related studies employ walking measures, such as walking time and walking distance, as outcomes. Walking time is an important indicator in patients with LSS and has been used as a measure in several studies [29,30]. In this study, walking time without pain was assessed at admission and long-term follow-up.

##### Survey at Long-Term Follow-up

In addition to the abovementioned outcome measures, operation status after discharge, treatment history for the last one month, treatment preference, level of satisfaction with the Korean medicine treatment, and PGIC scale score were surveyed at long-term follow-up. Examples of questions answered by patients were as follows: “Have you undergone a surgery after discharge?”, “Have you received the following treatments for the last one month?”, “How is your preference score of Korean medicine treatment?”, “How satisfied are you with the Korean medicine treatment you received during your hospital stay?”, and “How helpful is the inpatient treatment in terms of returning to work and adjustment to activities of daily living?” The PGIC scale evaluates improvement in patients in 7 levels, with the participants responding to a 7-point Likert scale for improvement in functional limitations after treatment (1 = very much improved, 2 = much improved, 3 = a little improved, 4 = no change, 5 = a little worse, 6 = much worse, 7 = very much worse). This assessment scale was originally developed for psychological purposes, but is currently used in other various fields of medicine to assess improvement in pain intensity [31].

### 2.5. Statistical Analysis

Statistical analysis was conducted on patients who responded to the long-term follow-up survey. Missing values were imputed using multiple imputation by predictive mean matching and the Markov chain Monte Carlo method. A total of 20 imputed datasets were generated, and sex and age were covariates for imputation.

With respect to the participants’ demographic characteristics, baseline outcomes, and long-term follow-up results, continuous variables were presented as mean and standard deviation, whereas categorical variables were expressed as frequency and percentage. Outcome values at each time point of admission, discharge, and long-term follow-up were presented as mean with 95% confidence interval (CI). A mixed-effects repeated-measures analysis was conducted to examine the change in outcomes from baseline to discharge and long-term follow-up. Time variables (admission, discharge, long-term follow-up) were included in categorical variables. Furthermore, 95% CI and *p*-value for the change in outcomes from baseline were presented.

In sensitivity analysis, the change in outcomes from baseline was analyzed using datasets that included non-respondents at long-term follow-up. A mixed-effects repeated-measures analysis was performed, and missing values were not imputed. All values were presented as 95% CI with *p*-value.

Survival analysis was conducted to model the time to improvement in main outcomes across the population. The minimal clinically important difference in main outcomes for LSS varies among previous studies (NRS score for back pain: 1.2–2; NRS score for leg pain: 1.25–1.6; ODI: 5–12.8 [32,33,34]). Hence, based on previous studies and an internal research team meeting, a decrease in the NRS score for back pain by ≥2 points, a decrease in the NRS score for leg pain by ≥2 points, and a decrease in ODI by ≥10 points were regarded as improvements in the present study. As the interval between measurements was long in this study, it was not possible to estimate the exact timing of improvement. In other words, the time to improvement was considered interval-censored between admission and discharge or between discharge and long-term follow-up. Therefore, nonparametric maximum-likelihood estimated survival curves were generated using the Expectation and Maximization-Iterative Convex Minorant algorithm for the interval-censored data (with ICLIFETEST procedures in SAS package [35]) [36]. The median time to improvement and 95% CI were presented.

## 3. Results

### 3.1. Participants

Overall, 759 out of 2212 patients admitted to four Korean medicine hospitals across the country with a diagnosis of LSS from 2015 to 2018 had MRI findings of central stenosis. Patients diagnosed with fractures and those without baseline values were excluded. Thus, the total number of eligible participants for long-term follow-up was 687. Among these patients, 378 patients were selected for analysis, excluding 309 patients who could not be reached or refused to participate in the study (Figure 1).

### 3.2. Basic Characteristics

The mean age of respondents at long-term follow-up was 62.21 ± 12.54 years; of these respondents, 251 (66.4%) were females. At the time of admission, the walking time without pain was 19.14 ± 19.05 min, and 65 (17.2%) patients exhibited neurogenic claudication symptoms. A total of 50 patients (13.2%) had previous spine surgeries, and the mean dural sac cross-sectional area (DSCA) was 58.26 ± 26.37 mm^2^ (Table 1). As for the non-respondents at long-term follow-up, the mean age was 62.28 ± 12.39 years, 453 (66.0%) were females, the walking time without pain was 18.76 ± 18.63 min, and the mean DSCA was 58.73 ± 26.46 mm^2^, which were all similar to the values of long-term follow-up respondents.

### 3.3. Intervention

The mean length of hospital stay was 23.48 ± 17.58 days (median: 21 days) among patients with LSS. Throughout the length of their hospital stay, patients received integrative Korean medicine treatments, including herbal medicine, acupuncture, electroacupuncture, pharmacopuncture, and Chuna manipulation (Appendix A).

### 3.4. Outcome Changes

The NRS score for back pain at admission was 5.73 points (95% CI, 5.58 to 5.87), which decreased by 2.06 points at discharge (95% CI, −2.23 to −1.9) and 2.20 points at long-term follow-up (95% CI, −2.41 to −1.99). At long-term follow-up, the NRS score for leg pain and ODI decreased by 2.28 points (95% CI, −2.59 to −1.96) and 17.31 points (95% CI, −19.6 to −15.02), respectively, whereas the walking time without pain and EQ-5D-5L score increased by 28.83 min (95% CI, 24.65 to 33.01) and 0.22 points (95% CI, 0.19 to 0.25), respectively (Table 2, Figure 2).

The analysis of 687 patients, including non-respondents, revealed similar results for outcome changes at long-term follow-up (i.e., a 2.19-point decrease in the NRS score for back pain (95% CI, −2.39 to −2.00), a 2.29-point decrease in the NRS score for leg pain (95% CI, −2.57 to −2.02), a decrease in ODI by 17.22 point (95% CI, −19.31 to −15.13), a 30.41-min increase in walking time (95% CI, 26.91 to 33.90), and a 0.23-point increase in the EQ-5D-5L score (95% CI, 0.20 to 0.25)) (Appendix A).

### 3.5. Survival Analysis

Survival analysis was conducted to model the time to improvement in the NRS score for back pain, NRS score for leg pain, and ODI across the population. The survival analysis indicated that the median time to improvement in the NRS score for back pain, NRS score for leg pain, and ODI was 9 days (95% CI, 8 to 10), 12 days (95% CI, 10 to 12), and 8 days (95% CI, 6 to 8), respectively (Table 3, Figure 3).

### 3.6. Survey at Long-Term Follow-Up

The median period from discharge to long-term follow-up was 1193 days (average: 1199.21 ± 392.75 days) (Table 4). A total of 210 (55.6%) patients responded that they were recommended for spine surgery prior to admission, whereas 38 (10.1%) patients reported that they underwent spine surgery after discharge. Furthermore, 65 (17.2%), 73 (19.3%), and 42 (11.1%) patients expressed that they were currently receiving Korean medicine treatment, Western medicine treatment, and both Western and Korean medicine treatments, respectively. With respect to the preference between Western and Korean medicine treatments, 264 (69.8%) patients responded that they preferred Korean medicine treatment, accounting for a higher ratio of preference, and the preference scores for Korean and Western medicine treatments (out of 10 points) were 7.78 ± 1.72 points and 4.90 ± 2.45 points, respectively. Additionally, 360 (95.4%) and 338 (89.4%) patients responded that their pain improved as compared to that at the time of admission and were satisfied with the Korean medicine treatment they received during their hospital stay, respectively.

## 4. Discussion

The present study showed that patients with LSS who received inpatient integrative Korean medicine treatment for an average duration of 23.48 ± 17.58 days exhibited significant improvement in the NRS scores for back and leg pain, ODI, and EQ-5D-5L score at discharge. Furthermore, the results from long-term follow-up confirmed that good prognosis was maintained among patients. The NRS score for back pain, NRS score for leg pain, and ODI decreased by 2.20 points (from 5.73 to 3.53 points), 2.28 points (from 4.78 to 2.51 points), and 17.31 points (from 45.72 to 28.41 points), respectively.

Based on the results of previous studies [32,33,34,37], the criteria for recovery in terms of NRS score for back pain, NRS score for leg pain, and ODI were conservatively regarded in this study as a decrease of 2 points, 2 points, and 10 points or more, respectively. According to these criteria, with respect to the NRS score for back pain, 57.1% and 61.4% of patients recovered at discharge and long-term follow-up, respectively, and the median time taken for recovery was 9 days. With respect to the NRS score for leg pain and ODI, nearly half of patients showed recovery at discharge, and intensive integrative Korean medicine treatment through admission facilitated rapid improvement in pain and functional disability.

The results of the long-term follow-up survey indicated that approximately 10% of patients underwent surgery after receiving integrative Korean medicine treatment and that more than half of patients were recommended for surgery prior to inpatient treatment, suggesting that LSS severity in these patients was not insignificant at the time of admission. Considering that LSS has a high estimated reoperation rate (approximately 7.2% at 1 year, 11.2% at 3 years [5], and 23% at 10 years [9]), a surgery rate of 10% within about 3 years (1193 days), which was the median follow-up duration in the present study, is therefore judged to be not high. Additionally, more than half of patients responded that they were not currently receiving treatment, indicating that several patients remained in good condition even without any treatment after approximately 3 years from the integrative Korean medicine treatment.

Most of the patients expressed that their current symptoms were improved as compared to those at the time of admission, and approximately 67% of patients responded that they had much improved. Furthermore, a lot of patients were satisfied with the Korean medicine treatment they received at admission and responded that it was helpful in terms of returning to activities of daily living and work (90% and 85%, respectively). The results of the questionnaire survey at long-term follow-up indicated that the level of satisfaction with inpatient integrative Korean medicine treatment was considerably high.

LSS is the most common cause of spine surgery among adults aged 65 years and older [3,4]. Nevertheless, in South Korea, which has a dichotomized system of Western and Korean medicine [11,12], several patients choose intensive inpatient integrative Korean medicine treatment for dorsalgia, including LSS [13]. The National Health Insurance Service’s 2018 National Health Insurance Statistics Yearbook revealed that approximately 65,000 individuals were admitted to Korean medicine hospitals for dorsalgia and other spondylopathies in 2018, which amounted to about one third of the number of those admitted to Western medicine hospitals [38]. Furthermore, patients tend to be more optimistic with respect to their expectation toward postoperative symptom improvement than toward actual surgical outcomes [39]. In some cases, patients even expect that they will fully recover from their conditions [40]. However, previous studies with follow-up after LSS surgery reported an ODI of 23–37 points, NRS score for back pain of 3–5 points, and NRS score for leg pain of 3–4.4 points at 1–2 postoperative years [33,34,41], indicating that pain and functional disability remain even after spine surgery. These findings and the results of the present study, which indicated a high level of satisfaction and symptom improvement after receiving integrative Korean medicine treatment, are expected to be helpful in describing the behavior of patients selecting inpatient Korean medicine treatment.

Meanwhile, according to predictor analysis not included in the paper, baseline outcomes, DSCA, previous lumbar spine surgery, neurogenic claudication, sex, and age were analyzed as significant predictors of the prognosis of patients with LSS who received integrative Korean medicine treatment (Appendix A). The odds of recovery were analyzed to be higher when baseline outcomes were worse, DSCA was wider, previous spine surgery experience was absent, and patients were males and younger. Neurogenic claudication status affected the result of the NRS score for leg pain only. The findings of this study support the results of previous studies [42,43,44,45]. In logistic regression models, the AUC was between 0.8 and 0.9. The explanatory power of a model is generally deemed to be good when the AUC is 0.8 or higher [46]. Therefore, it can be said that the predictive model of this study well explains the treatment prognosis of patients with LSS. Considering the factors included in the model may help explain the prognosis of treated patients.

In this study, the total number of eligible participants for long-term follow-up was 687; however, the number of respondents was 378 (response rate: 55.0%). This is believed to be largely influenced by the characteristic culture of South Korea, where cellphone numbers are frequently changed, and unknown numbers are not answered because of an experience of receiving various commercial and fraudulent calls. Nevertheless, the fact that the result of the analysis of outcome changes for participants, including non-respondents, shows no significant difference as compared to that for respondents. It only indicates that the low response rate has no considerable impact on the reliability of this study. Additionally, as a prospective observational study, this study has limitations in that it had no control group. Therefore, it could not determine the effects of Korean medicine treatment for LSS based on the study. However, the design of this study is consistent with the exploratory purpose of showing the progress of patients with LSS treated with Korean medicine as it is. Further, as physical therapy such as interferential current therapy, transcutaneous electrical nerve stimulation during hospital stay, and treatment after discharge were not evaluated in this study, the effects of such additional treatments cannot be ruled out. In the future, a well-organized registry study or a well-designed RCT comparing the effects of this treatment versus other treatment methods such as surgery will be needed.

This study is the first study to evaluate the long-term effects of integrative Korean medicine treatment on LSS. The results of this study with a long-term follow-up period of about 3 years on average confirmed that 378 patients with LSS who received inpatient integrative Korean medicine treatment were in good condition as compared to that at the time of admission. The Korean healthcare system is a dichotomized system characterized by the coexistence of Western and Korean medicine [11,12], and several patients are admitted to a Korean medicine hospital to receive intensive integrative Korean medicine treatment for conditions such as LSS or lumbar disc herniation [13]. Therefore, it can be said that South Korea has a suitable environment for the evaluation of the effects of intensive Korean medicine treatment. There exist no clear guidelines for LSS treatment method selection [8], and decisions concerning the treatment method are highly subjective and varied, depending on individual clinicians [47]. In these situations, the results of this study can serve as reference data for decision-making by clinicians who need to assess prognosis and determine the treatment method in clinical practice. Furthermore, this study highlights the potential of integrative Korean medicine treatment as a new treatment method for LSS.

## 5. Conclusions

The results of this study indicated an improvement in pain and functional disability among patients with LSS who received inpatient integrative Korean medicine treatment. The patients remained in good condition until long-term follow-up. The results of this study can be used as reference data for a physician’s decision on treatment methods and prognosis assessment. Further studies such as a well-designed RCT are needed to determine the effects of integrative Korean medicine treatment on LSS.

## Figures and Tables

**Figure 1 jcm-10-00074-f001:**
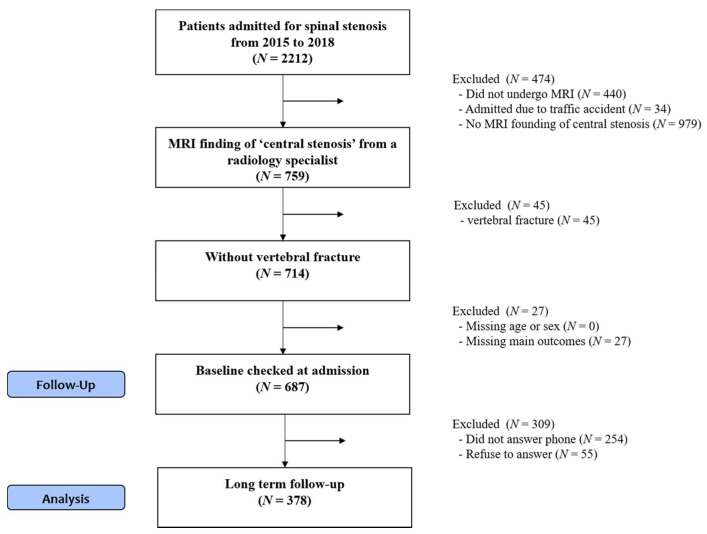
Flowchart illustrating participant enrollment.

**Figure 2 jcm-10-00074-f002:**
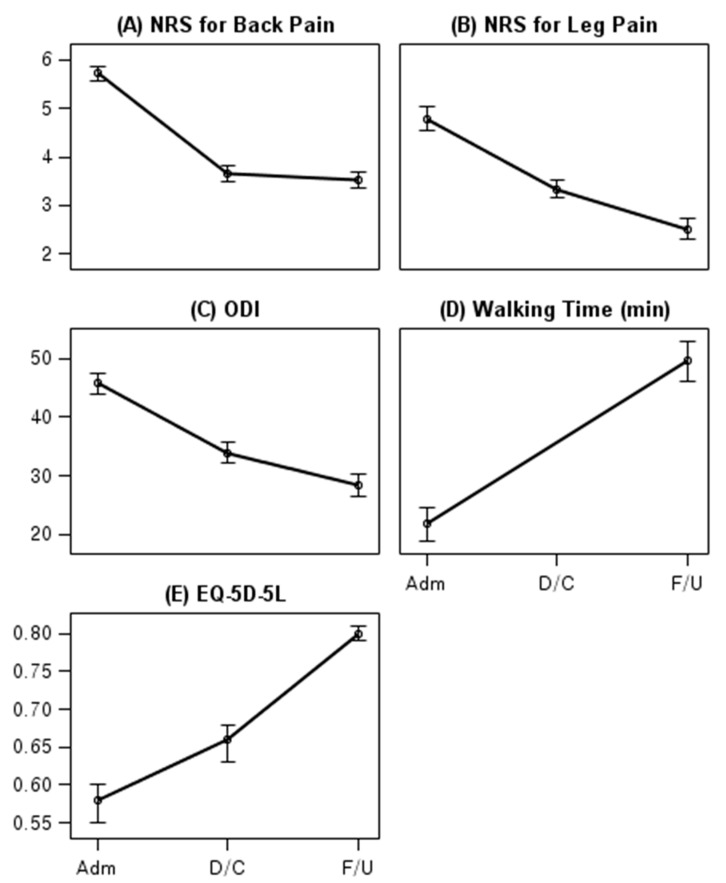
Changes in main outcomes at admission, discharge, and long-term follow-up. Outcome changes at baseline, discharge, and long-term follow-up are illustrated, with 95% confidence intervals represented by vertical bars. Plots and 95% confidence intervals were determined using a linear mixed model. (**A**) NRS for back pain on 11-point scale, (**B**) NRS for leg pain on 11-point scale, (**C**) ODI is presented as points, whereas (**D**) walking time without pain is presented in minutes (**E**) EQ-5D-5L to evaluate the participants’ quality of life is presented. NRS, numeric rating scale; ODI, Oswestry Disability Index; EQ-5D-5L, five-level EuroQol 5-dimension.

**Figure 3 jcm-10-00074-f003:**
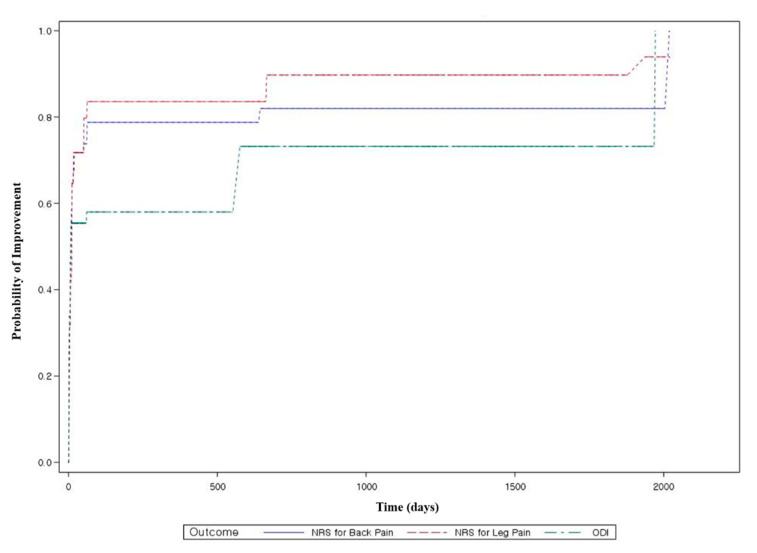
Nonparametric maximum-likelihood estimated survival curves using the Expectation and Maximization-Iterative Convex Minorant algorithm for the probability of improvement. Improvement was defined as a change of more than 2 points in the NRS scores for back and leg pain and of more than 10 points in ODI. The sections indicated with a dotted line are Turnbull intervals [35], which are sections in which the survival estimates cannot be uniquely determined. NRS, numeric rating scale; ODI, Oswestry Disability Index.

**Table 1 jcm-10-00074-t001:** Basic characteristics and clinical features at admission (*N* = 378).

	Values
**Age (years)**	62.21 ± 12.54
**Sex**	
Male	127 (33.6)
Female	251 (66.4)
**Smoking**	
No	318 (84.1)
Yes	57 (15.1)
**Drinking**	
No	287 (75.9)
Yes	84 (22.2)
**Body mass index (kg/m^2^)**	24.96 ± 7.96
**Pain radiating to the leg**	
None	69 (18.3)
Unilateral (right)	88 (23.3)
Unilateral (left)	101 (26.7)
Bilateral	120 (31.7)
**Walking time (min)**	19.14 ± 19.05
**Neurogenic claudication**	
Yes	65 (17.2)
No	184 (48.7)
**Muscle weakness**	
Big toe extension	
Normal	350 (92.6)
Weakness	28 (7.4)
Dorsiflexion	
Normal	352 (93.1)
Weakness	25 (6.6)
Plantar flexion	
Normal	363 (96.0)
Weakness	15 (4.0)
**Previous spine surgery**	
Yes	50 (13.2)
No	311 (82.3)
**NRS score for back pain**	5.73 ± 1.45
**NRS score for leg pain**	5.85 ± 1.30
**ODI**	45.72 ± 17.86
**DSCA (mm^2^)**	58.26 ± 26.37
**Accompanying HIVD**	115.0 (30.4)

Notes: Values are presented as frequency and percentage or as mean and standard deviation. Abbreviations: NRS, numeric rating scale; ODI, Oswestry Disability Index; DSCA, dural sac cross-sectional area; HIVD, herniated intervertebral disc.

**Table 2 jcm-10-00074-t002:** Changes in main outcomes at admission, discharge, and long-term follow-up.

	Admission	Discharge	Long-Term Follow-Up
NRS score for back pain			
Outcome	5.73 (5.58, 5.87)	3.66 (3.51, 3.82)	3.53 (3.35, 3.70)
Change from admission		−2.06 (−2.23, −1.9)	−2.20 (−2.41, −1.99)
*p*-value		<0.001	<0.001
NRS score for leg pain			
Outcome	4.78 (4.53, 5.04)	3.33 (3.14, 3.51)	2.51 (2.30, 2.72)
Change from admission		−1.46 (−1.69, −1.22)	−2.28 (−2.59, −1.96)
*p*-value		<0.001	<0.001
ODI			
Outcome	45.72 (43.91, 47.52)	33.94 (32.15, 35.73)	28.41 (26.46, 30.36)
Change from admission		−11.78 (−13.49, −10.07)	−17.31 (−19.6, −15.02)
*p*-value		<0.001	<0.001
Walking time (min)			
Outcome	20.72 (17.63, 23.81)		49.55 (46.12, 52.98)
Change from admission			28.83 (24.65, 33.01)
*p*-value			<0.001
EQ-5D-5L			
Outcome	0.58 (0.55, 0.60)	0.66 (0.63, 0.68)	0.80 (0.79, 0.81)
Change from admission		0.08 (0.05, 0.11)	0.22 (0.19, 0.25)
*p*-value		<0.001	<0.001

Notes: Missing values were imputed using multiple imputation. All values are presented as mean with 95% confidence interval. Linear mixed models were used to compute outcome changes and *p*-values. Abbreviations: NRS, numeric rating scale; ODI, Oswestry disability indes; EQ-5D-5, Five-Level EuroQol 5-dimension.

**Table 3 jcm-10-00074-t003:** Median time to improvement and percentage of improvement at discharge and long-term follow-up.

Outcome	Time (Days, 95% CI)	*N* (%) of Improvement	
		Discharge	Long-Term Follow-Up
NRS score for back pain	9 (8, 10)	216 (57.1)	232 (61.4)
NRS score for leg pain	12 (10, 12)	186 (49.2)	230 (60.8)
ODI	8 (6, 8)	176 (46.6)	215 (56.9)

Nonparametric maximum-likelihood estimated survival curves generated using the Expectation and Maximization-Iterative Convex Minorant algorithm were utilized to calculate the median time to improvement. Improvement was defined as a change of more than 2 points for the NRS scores for back and leg pain and of more than 10 points for ODI. NRS, numeric rating scale; ODI, Oswestry Disability Index.

**Table 4 jcm-10-00074-t004:** Results of long-term follow-up survey.

	Values
**Period from discharge to long-term follow-up (days)**	
Average	1199.21 ± 392.75
Median	1193 (839, 1488)
**Recommendation for surgery prior to admission**	210 (55.6)
**Experience of surgery after discharge**	
No	339 (89.9)
Yes	38 (10.1)
**Current treatment**	
None	198 (52.4)
KM	65 (17.2)
WM	73 (19.3)
KM + WM	42 (11.1)
**Preference**	
Preferred treatment	
KM	264 (69.8)
WM	19 (5.0)
Similar	95 (25.1)
Degree of preference	
KM	7.78 ± 1.72
WM	4.90 ± 2.45
**PGIC**	
Very much improved	92 (24.3)
Much improved	161 (42.6)
A little improved	107 (28.3)
No change	12 (3.2)
A little worse	6 (1.6)
Much worse	0 (0.0)
Very much worse	0 (0.0)
**Level of satisfaction with the KM treatment received during hospital stay**	
Very satisfied	155 (41.0)
Satisfied	183 (48.4)
Neutral	37 (9.8)
Dissatisfied	3 (0.8)
Very dissatisfied	0 (0.0)
**Degree of helpfulness of inpatient treatment** **in returning to activities of daily living and work**	
Very helpful	137 (36.3)
Helpful	186 (49.3)
Average	49 (13.0)
Not helpful	5 (1.3)
Not helpful at all	0 (0.0)

Notes: All values are presented are mean and standard deviation or as frequency and percentage. Median days are presented with 25% and 75% values. Abbreviations: KM, Korean medicine; WM, Western medicine; PGIC, Patients’ Global Impression of Change.

## Data Availability

The data presented in this study are available on request from the corresponding author. The data are not publicly available due to privacy/ethical restrictions.

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
