# Peer review of "Long-Term Follow-Up of Spinal Stenosis Inpatients Treated with Integrative Korean Medicine Treatment"

_jcm, 2020, doi:10.3390/jcm10010074_

Round 1

Reviewer 1 Report

The authors reported that patients with LSS who received inpatient integrative Korean medicine treatment exhibited an improvement in pain and functional disability at discharge.

The biggest issue is that the long-term effect is not able to be determined based on the data they presented, due to dozens of confounding factors (simultaneous use of Western medicine, physical therapy, etc..). 

As the authors concluded, the favorable effect at discharge still need to be evaluated by comparing with a control-group (RCT), otherwise the significance of this study is low.

Table 5 (multivariate analysis) is not a relevant data for the present study objective and seems redundant. Maybe should be deleted for the ease of readers. 

Minor comment:

Did the patients undergo any physical therapy during inpatient periods?

Line 298, 172% must be 17.2%?. 

Reviewer 2 Report

In this paper Kim et al report their experience with lumbar spinal stenosis treated with Integrative Korean Medicine.

It represents a relatively interesting paper, but in my opinion, it will be an interesting topic for a limited number of readers. 

Having said that, the paper is well structured, material and methods are well described, results seem reliable and the discussion is well organized.  

Author Response

Dear reviewer 2,

We appreciate your kind words for our manuscripts. We acknowledged that it would be an interesting work for a limited number of readers, but we hope it can provide useful information for that limited number of readers. We reconsidered the manuscripts and made several revisions.

We have moved predictor analysis to the supplementary part because we thought it to be a little redundant and have added some limitations in the discussion. We will continue to make efforts for further study, such as a well-organized registry study and RCT.

Round 2

Reviewer 1 Report

The authors have appropriately revised their manuscript.